# Numerous Trigger-like Interactions of Kinases/Protein Phosphatases in Human Skeletal Muscles Can Underlie Transient Processes in Activation of Signaling Pathways during Exercise

**DOI:** 10.3390/ijms241311223

**Published:** 2023-07-07

**Authors:** Alexander Yu. Vertyshev, Ilya R. Akberdin, Fedor A. Kolpakov

**Affiliations:** 1JSC “Sites-Tsentr”, 123182 Moscow, Russia; avertyshev@mail.ru; 2Department of Computational Biology, Scientific Center for Information Technologies and Artificial Intelligence, Sirius University of Science and Technology, 354340 Sochi, Russia; kolpakov.fa@talantiuspeh.ru; 3Biosoft.Ru, Ltd., 630058 Novosibirsk, Russia; 4Department of Natural Sciences, Novosibirsk State University, 630090 Novosibirsk, Russia; 5Federal Research Center for Information and Computational Technologies, 630090 Novosibirsk, Russia

**Keywords:** skeletal muscle, physical exercise, Ca^2+^-dependent signalling, AMPK signalling, protein phosphatases, mathematical model, transient process

## Abstract

Optimizing physical training regimens to increase muscle aerobic capacity requires an understanding of the internal processes that occur during exercise that initiate subsequent adaptation. During exercise, muscle cells undergo a series of metabolic events that trigger downstream signaling pathways and induce the expression of many genes in working muscle fibers. There are a number of studies that show the dependence of changes in the activity of AMP-activated protein kinase (AMPK), one of the mediators of cellular signaling pathways, on the duration and intensity of single exercises. The activity of various AMPK isoforms can change in different directions, increasing for some isoforms and decreasing for others, depending on the intensity and duration of the load. This review summarizes research data on changes in the activity of AMPK, Ca^2+^/calmodulin-dependent protein kinase II (CaMKII), and other components of the signaling pathways in skeletal muscles during exercise. Based on these data, we hypothesize that the observed changes in AMPK activity may be largely related to metabolic and signaling transients rather than exercise intensity per se. Probably, the main events associated with these transients occur at the beginning of the exercise in a time window of about 1–10 min. We hypothesize that these transients may be partly due to putative trigger-like kinase/protein phosphatase interactions regulated by feedback loops. In addition, numerous dynamically changing factors, such as [Ca^2+^], metabolite concentration, and reactive oxygen and nitrogen species (RONS), can shift the switching thresholds and change the states of these triggers, thereby affecting the activity of kinases (in particular, AMPK and CaMKII) and phosphatases. The review considers the putative molecular mechanisms underlying trigger-like interactions. The proposed hypothesis allows for a reinterpretation of the experimental data available in the literature as well as the generation of ideas to optimize future training regimens.

## 1. Introduction

Regular physical exercise initiates a number of adaptation processes in various systems of the human body. These exercises induce many metabolic and signaling events in skeletal muscle cells, which in turn activate downstream signaling pathways and induce the expression of many genes in skeletal muscle. One of the areas of research is mathematical modeling of metabolic and signaling processes (in order to test ideas about their mechanisms, simulate the response of signaling pathways under a wide range of loads, and then optimize training regimes). Based on models of metabolic processes [1,2], we developed a modular mathematical model in which exercise-induced metabolic processes are complemented by signal transduction and gene expression modules in human skeletal muscle [3]. The model includes Ca^2+^- and AMPK-dependent signaling pathways (Figure 1) and has been tested by modeling cyclic exercise on a bicycle ergometer and knee extension exercises of varying intensity.

The exercise has a significant impact on the factors affecting AMPK activity. Two primary factors, AMP and Ca^2+^ ion concentrations, are dependent on exercise intensity (force/power of contraction and frequency of dynamic contraction) and exercise duration. Therefore, exercise intensity and duration indirectly affect AMPK activity. This dependency has been demonstrated through experimental data and analyzed in detail in Section 2.

Increasing the exercise intensity results in a rise in the percentage of active muscle fibers, which can be attributed to increased force and/or fatigue. The metabolic changes occurring at the individual muscle fiber level are also influenced by the duration and intensity of the exercise. Elevated exercise intensity enhances the firing rates of motor neurons leading to a decrease in ATP and phosphocreatine (PCr) levels corresponding to the firing rates [4] and an increase in the concentration of calcium ions [5]. Together, these changes cause an increase in the concentration of AMP and enhance calcium/calmodulin-dependent protein kinase kinase 2 (CaMKK2) activity [6] within individual muscle fibers. The recruitment of additional muscle fibers and changes within individual muscle fibers, along with the duration of exercise, collectively lead to the observed elevation in AMPK activity at the whole muscle level.

The model simulation results are in agreement with the experimental measurements of metabolite concentrations and the activity of a number of kinases under continuous low (50% VO_2max_) and moderate (70% VO_2max_) aerobic exercise. However, when simulating intense and intermittent exercise, the model performs worse due to its limitations. These limitations are associated, among other things, with the simplification of the model by including mainly the most studied signaling pathways. Many details have been omitted, both due to the lack of relevant quantitative data and to provide an acceptable level of model abstraction (simplification). A number of improvements to the model are required to simulate a wider range of exercise regimens.

At the same time, it is not enough to complicate the model by simply adding the components of the metabolic and signaling pathways, since these components form much feedback. Adding such feedback to the model in many cases will lead to trigger-like interactions between elements of the signaling pathways and will require the search for quantitative data on the action of feedback loops. The data below suggests the existence of a number of transients during the activation of Ca^2+^- and AMP-dependent signaling pathways in skeletal muscle that may lead to unexpected responses, such as a decrease in the activity of certain AMPK isoforms in response to short-term exercise. These transients include changes in [Ca^2+^], [ADP], [AMP], and RONS concentrations and transients in the activity of various kinases and phosphatases. In this review, we summarize data from various studies and hypothesize that these transients depend on multiple feedback kinase/protein phosphatase interactions in human skeletal muscle. Since there are many reviews [7,8,9,10,11] describing numerous signaling pathways associated with exercise adaptation, we will not dwell on this topic in detail. Instead, we will focus on the relatively fast metabolic and signaling processes that may be responsible for the transients during the onset of exercise and consider possible molecular mechanisms underlying trigger-like interactions with a view to their subsequent modeling. In the future, this will allow for the creation of more adequate models and increasing their predictive potential.

The structure of the review corresponds to the analysis workflow. The starting point for the data analysis that led to the formulation of the hypothesis was the discrepancy between model and experimental data for interval exercise regimens. A comparative analysis of the data from several studies, detailed in Section 2, suggested that the observed dynamics of AMPK and CaMKII activity in the time window of 0–10 min from the onset of exercise can reflect the course of transient processes. Since AMPK and CaMKII activities are regulated by phosphatases along with [AMP] and [Ca^2+^] as primary regulators, we searched for and analyzed data on the regulation of phosphatase activity, detailed in Section 3. We then identified additional factors that markedly change during exercises and significantly affect the activity of kinases and phosphatases. These factors are described in Section 4. Based on the data analysis, we formulate the hypothesis that the observed changes in AMPK activity may be largely related to metabolic and signaling transients rather than exercise intensity per se and try to interpret the experimental data using this hypothesis. Furthermore, we speculate how these hypothetical dependencies might be influenced by muscle fiber types, muscle fiber recruitment, metabolite concentration dynamics, and fatigue. The hypothesis and possible interpretation of the experimental data are described in Section 5. At the next stage, we subject the generalized hypothesis to criticism in order to identify additional weaknesses. We took into account the substrate specificity of various isoforms of kinases and phosphatases, on which the expected feedback depends. The identified problems are described in Section 6. Overall, the discussion built around the proposed hypothesis focuses on the remaining data gaps that hinder the further development of dynamic models.

## 2. AMPK and Ca^2+^—Dependent Signaling Transients

To search for possible reasons underlying the discrepancies between simulations and experimental data, we compared available data from studies that considered exercises of varying intensity and duration. As a result, it was suggested that the observed dynamics of AMPK and CaMKII activity at the beginning of exercise reflect the course of transient processes. In this section, we describe the initial dataset, the observed peculiarities, and the shortcomings of our simulation.

### 2.1. Dynamics of AMPK and CaMKII Activity during Exercise

During the onset of exercises, AMPK isoforms demonstrate different activities. In general, changes in AMPK isoform activity depend on exercise intensity and duration. A muscle biopsy is used to determine AMPK activity in vivo. Typically, a biopsy is taken at a limited number of time points due to the complexity of this invasive procedure. For example, samples were taken six times per participant (before and during training) [12], but usually there are even fewer samples. Since the minimum time required for a biopsy procedure is about 15–30 s, this also imposes limitations on the exercise regimens that can be studied (including interval exercises). After the biopsy, a complex multistage procedure [13] is performed to measure the activity of the obtained AMPK and its isoforms using various substrates.

While α2 AMPK, especially α2β2γ3 isoform activity, commonly rises after exercise at intensities above 50% VO_2peak_ and higher [12,14,15,16], the α1 AMPK activity changes are controversial. The controversies in the data from different studies may be partly due to the different training statuses of participants and different exercise intensities. In untrained subjects, α1 AMPK activity increases or tends to increase [17,18,19], while the activity of α1 AMPK during exercise does not change or tends to decrease in trained subjects [12,14,15,16,20,21,22,23]. One study even showed a decrease of α1 AMPK activity after high-intensity all-out sprint exercise [16] (Figure 2). These data are discussed in detail in another review [24].

Among the cited works [14,15,18,19,20] used the SAMS peptide, while [12,16,22,23] used the AMARA peptide. In various studies, the amount of lysate used in measurements varied from 50 to 300 μg. To overcome methodological differences in these studies, we converted AMPK activity data from different studies into a single FoldChange indicator.

However, we noticed some peculiarities in the change in AMPK activity in response to the exercise of analyzing the published data. If the time points at which AMPK activity was measured were far apart (10 min or more) then a progressive increase in the activity was observed as a rule (Figure 3).

If the biopsy was taken more often, then signs of transient processes would be observed. At the beginning of the exercise, a downward trend in the activity of α1β2γ1 and α2β2γ1 isoforms of AMPK was observed [12], as well as a slow rise in α2β2γ3 activity. Then, the AMPK activity increased during the progression of exercise (Figure 4).

Therefore, the decrease in AMPK activity after high-intensity exercise observed in a number of studies, including [16], may not be a fundamental dependence, but represent only part of a transient process.

An opposite transition but fairly similar trend was observed for CaMKII phosphorylation and autonomous activity during exercise on a bicycle ergometer at intensity corresponding to 67 ± 2% of VO_2peak_ for 90 min [27]. In this study, biopsies were taken from the *vastus lateralis* muscle before and after 1, 10, 30, 60, and 90 min of exercise. At the 1 min time point, there was a significant increase in CaMKII phosphorylation and autonomous activity, which decreased significantly at the time point of 10 min and beyond (Figure 5). The trained subjects (VO_2peak_ 55 mL/kg/min) participated in this study [27], just like in the [12] study (VO_2peak_ 55 mL/kg/min).

The transients in CaMKII autonomous activity may be related to differences in Ca^2+^ uptake and release kinetics and thus Ca^2+^–CaM binding, or changes in the activity of protein phosphatases dephosphorylating CaMKII over time [27].

It should be noted that assumptions about transient processes were made using the data obtained in mixed fiber muscle extracts (homogenate and lysate). Individual muscle fibers of various types and motor units may have significant differences in the concentration of metabolic enzymes, kinases, and phosphatases. During exercise of different intensities, the concentrations of metabolites in different muscle fibers can vary greatly, both due to uneven recruitment and differences in fiber types. We assume that the amplitudes of transients at the level of individual fibers of various types and motor units may be much higher compared to those measured in homogenates not separated by fiber types, and the changes may reach statistical significance.

Taken together, the data suggests that some transient processes occur in a short period of time (1–10 min from exercise onset), the underlying causes of which are not yet clear. Unfortunately, most of the studies discussed above were performed using continuous exercises with constant loads of different intensities. There is no data on what transients exist during variable intensity training and competitions. For example, during various interval training exercises or when moving along a course with uphills, downhills, and flat terrains. 

### 2.2. Possible Effects of Repeated Exercise

It is known that during repeated bouts of physical exercise, different kinetics of oxygen consumption [28,29,30,31] as well as different rates of activation of a number of metabolic enzymes [32,33,34] and, respectively, different kinetics of metabolic processes (ON-kinetics) [28,35,36,37] are observed.

Unfortunately, the kinetics of the activity of metabolic enzymes and kinases in the pause between exercise bouts or after exercise cessation (OFF kinetics) have not been comprehensively studied. To our knowledge, there are no in vivo investigations with data on OFF-kinetics with sufficient time resolution, although the data are very important since they can additionally characterize and support a hypothesis on the mechanism of transient processes. Such processes can play an essential role in actual training practice, determine the effectiveness of various training programs and exercise regimens, and provide insight into how to optimize training regimens.

### 2.3. Simulations Based on the Current Model and Their Limitations

Our current model by [3] simulates a non-linear increase in AMPK phosphorylation as a surrogate for its activity (ON-kinetics) during moderate-intensity exercise (Figure 6B,C), which is similar to the increase in AMPK activity that was proposed by [24] based on experimental data (Figure 6A).

The simulated AMPK activity during recovery shows an almost exponential decline in the model [3]. At the same time, there is a study demonstrating that the decrease in AMPK activity occurs with some delay [38]. The decrease of AMPK activity in vitro takes a sigmoid-like shape after the addition of protein phosphatase with about a 10-minute delay (Figure 7). The reasons for such a delay are unexplained, though. A rather similar pattern of AMPK activity in vitro is shown by [39], but the sigmoid-like shape is almost indiscernible, probably due to different experimental conditions and a much higher rate of dephosphorylation.

The current set of model equations [3] cannot simulate changes in AMPK activity different from exponential-based rise and drop dynamics (Figure 6B,C) and particularly transient processes. To adequately simulate the transient processes, it is necessary to find out the underlying causes and incorporate them into the model. Therefore, we had to search for additional quantitative and qualitative data.

One of the directions was the interrogation for data on the kinetics of protein phosphatase activity during exercise (ON and OFF phases). Since it is necessary to clarify the shape of the decrease in the activity of AMPK and other kinases after exercise and in pauses of 1–10 min during interval training.

## 3. Trigger-like Kinase/Protein Phosphatase Interaction Networks in Central Nervous System (CNS)

First, we performed an extensive search for data on the structure of protein phosphatases, their functions, and their regulation in various tissues and cell lines in order to select those that are relevant to our task. We intended to take the kinetic data from studies conducted on muscle tissue, preferably in humans. Most studies related to the regulation of protein phosphatases have been performed on cells of the nervous system. We hypothesize that some of the described dependencies might also exist in muscle cells and consider the possibility of modeling them, taking into account the differences between muscle cells and neurons.

### 3.1. Regulation of Protein Phosphatases in the CNS

We performed an analysis of the published data on the regulation of protein phosphatase activity, starting with reviews [40,41,42,43,44]. The identified regulatory factors were divided into several groups according to their degree of relevance to the issue:Regulators of the expression of protein phosphatases and regulatory proteins; modulators of the assembly of protein phosphatase complexes (long-term processes, hours, days);Regulators of post-translational modifications associated with long-term processes; pathways related to hormones and nutrients (minutes, hours);Regulators of activity and post-translational modifications that have a higher speed of action, comparable to the speed of processes during the onset and cessation of the exercise (seconds, minutes).

The factors from the third group can be considered candidates for the model extension. The remaining factors can be considered as determining the constitutive activity of protein phosphatases at rest and can be taken into account as the model parameters. An approximate distribution of protein phosphatase regulatory factors into three groups with examples is shown in Figure 8.

Most of the articles found and related to the studies of fast regulation were performed on cells of the nervous system. Since muscles are activated by neural impulses and muscle cells change their internal state in response to each impulse (or burst of impulses), we assumed that the rates of similar processes in neurons and muscles are comparable. And the data from studies on neurons can be extrapolated to some extent to the skeletal muscles.

Studies on various brain cells have revealed rapid opposite changes in the activity of kinases and protein phosphatases, which can be considered trigger-like interactions. A number of computational models were developed for various signaling cascades [45,46,47,48].

Studies of long-term depression (LTD) and long-term potentiation (LTP) show that on/off processes are based on feedback loops [48], including CaMKII, PP2A, and PP1, and other participants, such as the protein kinase C (PKC) and mitogen-activated protein kinase (MAPK) positive feedback loop [47]. Interestingly, the time courses of CaMKII and PP2A activity changes [47,48] are the same as for the skeletal muscle transients described above. PP2A activity suppresses the feedback loop, while inhibition of phosphatase activity by CaMKII disinhibits the feedback loop. Additionally, nitric oxide (NO) facilitates activation of the positive feedback loop by activating the cGMP/protein kinase G (PKG) pathway, which supports phosphatase inhibition [48]. It is suggested that active CaMKII indirectly inhibits PP1 and PP2A by the phosphodiesterase 1 (PDE1)-PKG signaling pathway [47,48]. Direct inhibition is also potential as CaMKII phosphorylates PP2A regulatory subunit B56α, which leads to a decrease in PP2A activity [49]. The feedforward and feedback loops in a signaling network contribute to the nonlinear response. As a result, PP1 activity decreases along with an increase in CaMKII activity [46]. While PP2A is the major enzyme that dephosphorylates CaMKII and reduces its activity [50], the existence of positive feedback is also plausible. The active PP2A-B55β isoform dephosphorylates specific CaMKII sites (Thr305, Thr306, and Ser314) and slightly increases CaMKII activity in response to increased nutrient concentrations [51].

As a result of CaMKII, protein phosphatases PP2A and PP1, and additional feedback interactions, a «trigger» network is formed. The question arises from the data: how do primary stimuli regulate the switch of such a trigger? CaMKII activity switches from very low basal activity to complete activity within a very narrow calcium concentration range [52,53]. Based on the numerous experimental data, the switch of such a trigger in neurons was modeled [45]. It has been shown that the trigger is switched mainly by changes in Ca^2+^ concentration (Figure 9).

### 3.2. Are Similar Regulation Loops Possible in Skeletal Muscles?

Assumptions about a similar trigger-like system in the skeletal muscle should be made with caution due to differences in kinase and phosphatase concentrations in the brain and skeletal muscle. CAMKII is one of the most abundant proteins in all brain structures and makes up about 0.6% of hippocampal proteins, 0.55% of cortical proteins, and about 0.2% of cerebellar proteins [54,55]. Calculated estimates of kinase and phosphatase concentrations in skeletal muscle [3] based on the data from [56] show CaMKII concentrations about one to two orders of magnitude lower than in brain structures. Despite the large difference in concentrations, the [CaMKII]:[PP2A] ratio in the brain and skeletal muscle is comparable; for this evaluation, concentration calculations for the model Akberdin et al., (2021) [3] were used. These calculations were based on human muscle proteomic data provided by Murgia et al., (2017) [56]. So, we propose that similar trigger-like interactions can contribute to transient processes in skeletal muscles as well.

In the context of skeletal muscle modeling, one can omit some CNS-specific feedback loops and consider mainly the effect of [Ca^2+^] and muscle-specific factors. The concentration of calcium during muscle contraction is proportional to the frequency of impulses [5], and frequency can change several times upon activation of the muscle fiber [57,58,59]. It can be assumed that the multistable CaMKII, PP1, and PP2A systems can also exist in skeletal muscles, and their switch depends on the impulse rate and duration, as observed in the CNS [48]. Since the concentration of calcium in the muscle fiber during muscle contraction varies significantly in different compartments [60], [Ca^2+^] near the sarcoplasmic reticulum (SR) can be comparable to the concentrations described in [45] for the CNS and cause switching of the trigger system. Perhaps, [Ca^2+^] threshold in skeletal muscle fibers is different from the threshold in neurons due to differences in CaMKII, PP1, and PP2A concentrations and their localization.

In situ stimulation activates or deactivates various CaMKII isoforms in mouse muscle [61] depending on the frequency of stimulation, which may to some extent indicate that this is due to the trigger-like switching. Although the opposite effect is shown for 𝛿A-CaMKII compared to neurons (possibly due to the specificity of different PP isoforms to different substrates).

After the cessation of the exercise, Ca^2+^ concentration drops almost instantly. The subsequent trigger state is determined by the different rates of decrease in the activity of CaMKII, PP1, and PP2A, and, likely, some other components of the trigger system.

In addition to the potential CaMKII/(PP1,PP2A) trigger, there is another possible trigger represented by the AMPK/PP2A pair in human skeletal muscles. It is known that PP2A dephosphorylates AMPK, reducing its activity. At the same time, it was shown that AMPK can activate PP2A [62]. Thus, this additional AMPK/PP2A feedback loop can work in conjunction with the CaMKII/(PP1,PP2A) trigger.

Many similar triggers probably exist since this is a fairly common phenomenon. For example, the association between protein kinase B (AKT) and AMPK activity (pAkthigh/pAMPKlow and pAMPKhigh/pAktlow) in cancer cells is trigger-like [63]. Another example is that AMPK phosphorylates heat shock factor 1 (HSF1) and reduces its activity; in turn, HSF1 can suppress AMPK activity [64].

## 4. Additional Regulatory Factors in Skeletal Muscles

Many additional factors may influence the regulation of AMPK, CaMKII, and protein phosphatases in skeletal muscle. We selected and reviewed those that meet the criteria: they change significantly during exercise and significantly affect the activity of kinases and phosphatases. Along with [AMP] and [Ca^2+^], we considered the following factors as regulators of AMPK and CaMKII activity during single or intermittent exercise:

### 4.1. Reactive Oxygen and Nitrogen Species

Reactive oxygen and nitrogen species (RONS) may activate or deactivate AMPK by modifying RONS-sensitive residues of the AMPK-α subunit (Figure 10). RONS may activate AMPK by reducing mitochondrial ATP synthesis and subsequent AMPK phosphorylation by liver kinase B (LKB1) and CaMKKβ. RONS reduces the rate of Thr172-AMPK dephosphorylation [65]. Alternatively, RONS modifies the Met281/282 pair within the regulatory domain of CaMKII and activates it via a parallel mechanism to Thr286 autophosphorylation [66]. H_2_O_2_ induction of ROS inhibits PP2A activity [67]. A number of RONS can inhibit or activate PP2A [43]. Antioxidant ingestion before exercise reduces AMPKα phosphorylation by reducing Thr286-CaMKII and increasing Ser485-AMPKα1/Ser491-AMPKα2 phosphorylation in human skeletal muscle and results in lower AMPKα and CaMKII activities [68]. Too high or too low RONS levels may lead to the inhibition of Thr172-AMPKα phosphorylation [65].

### 4.2. Muscle Glycogen

During exercise in the glycogen-depleted state, AMPK phosphorylation and activity rise significantly higher than in the glycogen-loaded state [26,69] (Figure 10). This rise in activity in a glycogen-depleted state is accompanied by reductions in α1 and α2 AMPK association with glycogen, along with α2 AMPK translocation to the nucleus following exercise [70]. As a consequence, exercise in a glycogen-depleted state induces significantly higher peroxisome proliferator activated receptor (PPAR) gamma coactivator-1 alpha (PGC-1α) mRNA expression [71]. More detailed information on the relationship between glycogen availability and AMPK response to exercise is represented in the review [72]. On the other hand, the reduction in muscle glycogen causes a decrease in SR Ca^2+^ release rate and uptake rate [73,74,75], thus affecting [Ca^2+^] and Ca^2+^/CaM-stimulated CaMKII activity [76,77,78] and other Ca^2+^-dependent signaling pathways.

### 4.3. Insulin

AMPK activity is regulated by insulin and insulin-like growth factor 1 (IGF-1) mainly through the activation of Akt and phosphorylation of α-AMPK Ser485/491, which reduces AMPK activity [79]. Simultaneously increased levels of insulin enhance PP1 activity and reduce PP2A activity [80,81,82,83] (Figure 10). Plasma insulin shows a marked and sharp rise during recovery after a bout of intense exercise due to rapid perturbations of glucose production and utilization [84]. Therefore, insulin may play a role in the regulation of AMPK activity during rest periods of interval training regimens. The concentration of insulin depends on the fed or fasted state of the participants. The time course, magnitude, and direction (increase or decrease) of the kinase activity are different depending on the fed and fasted states before the exercise [85]. These data emphasize the importance of nutrients and insulin in the post-exercise response of signaling pathways and highlight the problems of applying fasted state research findings to real-life training, as exercise in a fasted state is not a common practice.

### 4.4. Adrenaline

Exercise induces an increase in adrenaline and noradrenaline concentrations in the blood, and this increase depends on the duration, intensity, and type of exercise. Stimulation of β-adrenergic receptors activates the cAMP signaling cascade and may indirectly promote AMPK activity, as demonstrated by [86]. Increasing the concentration of cAMP during β-adrenergic receptor stimulation may lead to the simultaneous elevation of AMP, a degradation product of cAMP resulting from PDE activity. As a result, the AMP:ATP ratio increases and promotes AMPK activity. The inhibition of the adenyl cyclase in mouse extensor digitorum longus (EDL) muscles as the inhibition of β-adrenergic receptor signalling by propranolol prevents increasing the concentration of cAMP, increasing the AMP:ATP ratio, and increasing AMPK activity [87]. PP2A activity is also regulated by β-adrenergic receptor stimulation through cAMP-protein kinase A (PKA) pathway-mediated phosphorylation of the PP2A regulatory subunit, which increases PP2A activity [88,89,90] (Figure 10).

The regulation of AMPK and CaMKII activities is very complex and involves a number of feedback loops. Not all of them are discussed above. Many known minor potential pathways and actors of a plausible trigger system are omitted, especially long-term factors not relevant to fast, transient processes. The described factors: RONS, muscle glycogen, adrenaline [91], and insulin [84] change during and after exercise, so they may shift thresholds for potential triggers during exercise and rest intervals.

## 5. Hypothesis

Assuming the presence of transient processes described in Section 2 and the possible similarity of feedback loops in the interaction of kinases/phosphatases in the CNS and muscles described in Section 3, we considered them collectively.

### 5.1. Generalized Scheme

Here we summarize the data of the studies discussed above and offer a generalized view of the trigger system in the first approximation (Figure 10). This scheme focuses on PP2A due to its known interactions with AMPK, CaMKII, Ca^2+^, RONS, and several other factors.

Based on numerous data on skeletal muscle signaling during exercise and substantial contributing factors, we hypothesize that:Observed decreases in activity of α1β2γ1 and α2β2γ1 isoforms of AMPK and delayed increases in activity of α2β2γ3 isoforms at exercise intensity 70% of VO_2max_ and above [12,16] are determined by coupled metabolic and signaling transient processes at the onset of exercise in a time window of about 1–10 min;These transients may in part be due to putative trigger-like kinase/protein phosphatase interactions similar to the nature of interactions in the CNS;Numerous dynamically changing factors: [Ca^2+^], metabolite concentration, RONS, and intramuscular glycogen hormones, can shift switching thresholds and change the state of «triggers», thereby affecting the kinase activity.

This generalized hypothesis can be applied to interpret some of the observed results in exercise studies. Herein, we give an example of such data interpretation. One of the results we wanted to simulate was [92]. In this study, signaling data was measured and compared across two exercise regimens (Figure 11).

As shown in Figure 11, intermittent exercise induces higher phosphorylation of AMPK and CaMKII compared to continuous exercise. One of the reasons may be insufficient time for a continuous trial, as 30 min of moderate-intensity exercise did not affect AMPKα1/2 Thr172 phosphorylation or induce PGC-1α expression in trained athletes [93]. As sedentary individuals take part in [92] (VO_2peak_: 44 ± 6 mL/min/kg), other factors may prevail.

Based on the fast drop of CaMKII phosphorylation and activity after the fast initial rise during exercise at the same intensity corresponding to 67 ± 2% of VO_2peak_ [27], we speculate that suppressed AMPK activity after a continuous trial [92] is linked to suppressed CaMKII phosphorylation and its activity and to increased PP2A activity (trigger-like switch), respectively. At the same time, CaMKII phosphorylation and activity have not likely declined sharply during intermittent exercise, and CaMKII was able to suppress PP2A activity. Thereby, phosphorylation of AMPK and p38-MAPK has also increased after intermittent trials.

It can also be hypothesized that at the beginning of exercise there is a sharp increase in ROS production above the optimal level, which leads to a decrease in AMPK activity [65], and then, probably with some delay in time, there is an increase in the activity of superoxide dismutase (SOD) and catalase, and the concentration of ROS falls below the inhibitory level. Various assumptions can be made about the rest of the components of the scheme of potential triggers (Figure 10). There is no data on the dynamics of such processes with sufficient time resolution to correspond to the intermittent exercise given in the example. So these potential assumptions can be underlying causes as well.

If the area under the curves of the AMPK, CaMKII, and p38-MAPK activities is interpreted as an indirect measure of the subsequent activation of gene expression leading to more protein production, then it can be assumed that this intermittent exercise regimen will be more effective. Therefore, plausible trigger-like interactions may play a significant role in the effectiveness of various exercise modalities.

### 5.2. View at the Level of the Individual Muscle Fibre and Fibre Type

As mentioned above, the data of the majority of studies were obtained from mixed fiber muscle extracts (homogenate and lysate) [12,16,27,92] and others referred to in the text. Thereby, the amplitudes of transient processes at the level of individual fibers of different types and motor units can be much higher or lower. Thus, fiber types and recruitment patterns should be considered in the context of triggers.

[Ca^2+^] is one of the main regulators of CaMKII activity and indirectly of AMPK activity. The concentration of calcium during muscle contraction is proportional to the frequency of impulses [5], and this frequency can change several times upon activation of the muscle fibers of the motor unit. Impulse frequency during motor unit activation is different for low and high threshold motor units [57,58,59] and has a different pattern of frequency increase during muscle contraction while intensity increases. Thus, tetanic [Ca^2+^] depends on the recruiting pattern and firing rate for a given motor unit. On the other hand, an overall decline in the mean motor unit firing rates during repeated maximal [94] and submaximal [95] contractions was observed. During fatigue-induced protocols, tetanic [Ca^2+^] also decreases [5,96]. This [Ca^2+^] decrease relies on subcellular glycogen depletion [96]. Thus, a decline in mean motor unit firing rates and a decrease in [Ca^2+^] release rate can both affect trigger threshold.

The single muscle fibers of different types have different concentrations of protein kinases and protein phosphatases, as well as their isoforms [16,19,22,56,97]. These concentrations change as a result of training [19,98,99] depending on the training status as well as on genetic determinants. The concentration of the AMPK α2β2γ3 heterotrimer, the most sensitive to exercise isoform, substantially decreases after training. As a result, the overall response of AMPK to exercise is blunted at high training levels [19]. Since the activity of metabolic enzymes, protein kinases, and protein phosphatases is related to their concentration, trigger switching thresholds may depend on the fiber type and training status. The activity of metabolic enzymes among muscle fibers of one motor unit is very close [100,101]. Therefore, it is sufficient to simulate [Ca^2+^], the activities of metabolic enzymes and kinases, and transients at the “conditional” motor unit level.

As well as trigger switching thresholds, which may depend on the concentration and activity of metabolic enzymes, protein kinases, and protein phosphatases, these thresholds also depend on the dynamics of metabolite concentrations. During high-intensity exercise, especially when exercising to failure, there is no steady state in vivo. There is a progressive change in the concentrations of ATP, ADP, Pi, creatine phosphate, creatine, and other metabolites, and the pH decrease in the cytosol continues [102,103]. At the cellular level, a decrease in pH leads to a deterioration in the interaction of actin and myosin, and together with an increase in [P_i_] leads to a decrease in the single fiber force [104,105,106]. A decrease in pH leads to a decrease in the rate of oxidative phosphorylation [107,108,109,110]. The rate of glycolysis also decreases along with the pH drop [111,112]. As a result, the concentration of ATP decreases, and consequently, the AMP:ATP ratio changes, affecting AMPK activity. Also, a change in the concentration of ROS and a decrease in pH impact the buffering of calcium in mitochondria [113] and its subsequent release, thereby affecting the dynamics of calcium concentration in the cytosol [114]. These dynamics may play a role during interval training exercises, although the extent of their influence is not clear.

Summarizing, the interaction between kinases and phosphatases depends on their concentrations, metabolite levels, and various other factors. These factors are influenced by fiber type and training status. Consequently, the effect of the same exercise may vary for different athletes at different stages of training. This could potentially explain the wide variations observed in experimental data, even among relatively homogeneous groups of subjects.

## 6. Issues

Since the generalized hypothesis is based on too little experimental data, in the next step we subjected this hypothesis to criticism in order to identify additional limitations, weaknesses, and missed factors.

Ideally, the model should take into account many of the factors described above and additional factors. Such as localization of organelles, kinases, and metabolic enzymes in compartments of muscle cells, restriction of diffusion of metabolites, and proteins that change their localization [115,116,117]. Unfortunately, we are faced with a lack of sufficient experimental data in most cases. Conducted studies have identified the presence of a phenomenon but often do not provide sufficient data to determine the relative “weight” of the phenomenon or mechanistically model it. Therefore, it seems more realistic to test the concept of triggers on a simplified model, even modeling not the muscle as a whole but “conditioned” motor units of different types, varying the initial variable values and parameter values (including the concentrations of metabolites and components of signaling pathways).

There is another source of issues in constructing a hypothesis. The presented general scheme (Figure 10) is an “optimistic” version. The next step is to take into account factors like the substrate specificity of kinases and protein phosphatases. Available data from a number of studies is summarized in Appendix A. Applying these data to the scheme reveals a number of gaps and blind spots (Figure 12).

Phosphoproteome screening studies often show conflicting results between in vivo and in vitro studies with various purified PP2A isoforms. For example, although PP2A B55δ (Ppp2r2d) regulates the phosphorylation state of AMP kinase by dephosphorylating Thr-172 [125], phosphoproteomic screenings [126,127] did not confirm this result. These contradictory results may be due to differences in the cell lines used, methods, etc. Thus, non-significant interactions from phosphoproteomic studies can also be interpreted with certain assumptions as possible interactions. The type of interaction, activation or inhibition of PP2A, depends on phosphorylated sites. While double-stranded RNA-activated protein kinase (PKR) wild-type B56α phosphorylated at multiple sites increases the activity [118], B56α phosphorylated at Ser41 decreases the activity [123].

The scheme of isoform-specific interactions (Figure 12) shows the absence of many relationships potentially arising from the generalized scheme (Figure 10). For example, it follows from the generalized scheme that the protein phosphatase PP2A dephosphorylates AMPK, and AMPK, in turn, activates PP2A, forming two feedback loops that lead to a potentially unstable state of the AMPK-PP2A system. If isoform-specific interactions are taken into account, the available data show that different PP2A isoforms are involved in these loops, and the overall loop becomes open, which casts doubt on the existence of a simple trigger. Perhaps these loops are more complex and include long chains similar to the CAMKII-PDE1-PKG-PP2A chain described for the CAMKII-PP2A system by [47], along with the direct inhibition of the PP2A B55α isoform by CAMKII found in [49]. There is little data on direct isoform-specific interactions for the CAMKII-PP2A system to reasonably speak about their participation in a trigger-like system. PP2A activation by Ca^2+^ ions is also selective; activation of only the PR72α (Ppp2r3a) isoform was confirmed by [120]. This makes the generalized scheme (Figure 10) questionable. Thus, it is difficult to prove or disprove the initial hypothesis. This scheme also does not include processes of autophosphorylation and autodephosphorylation of kinases or protein phosphatases [128,129]. Such processes may partly explain in vitro observations (that exclude most in vivo proteins and complex interactions), such as the unexplained sigmoid-shape decrease in AMPK activity [38] and a sigmoid-shape decrease in PP2A activity [128].

The proposed generalized and isoform-specific schemes (Figure 10 and Figure 12) do not include PP1. PP1 and PP2A are the major protein phosphatases in skeletal muscle [40] and act in concert despite their different substrate preferences and regulation [130]. PP1 regulates glycogen metabolism and contractile function in skeletal muscle [131,132,133], like many other processes. There is evidence that PP1 can regulate PP2A activity [134,135], so PP1 could potentially be included in the proposed trigger-like model for skeletal muscles. 

Although the full isoform-specific interactome remains to be studied, existing models [46,47,48] show that simulation and validation can be performed without taking such isoform-specific interactions into account. More detailed models can be developed as soon as the necessary data becomes available.

## 7. Conclusions

The considered examples indicate that at the beginning of exercises, after their cessation, or in pauses during interval training exercises, rather fast transient processes take place, affecting the activity of protein kinases and protein phosphatases. A preliminary analysis of the data represented in the numerous articles allowed us to propose a hypothetical scheme that determines these transients.

In general, the concept of triggers that switch based on [Ca^2+^] and metabolite concentrations (and other signals) could explain many nonlinear signaling events during and after physical exercise. Probably, some local feedback loops lead to a trigger-like switch between stable states, while others lead to a shift in the balance of kinase/phosphatase activity and unstable equilibrium. The last ones, in turn, can affect the switching of trigger-like subsystems. Taken together, these interactions may lead to the observed dynamics of AMPK and CAMKII activity. In view of the limited relevant data, the hypothesis becomes an intriguing challenge for mathematical modeling. If variations of the model parameters could reproduce some of the observed transients, then a potential area of trigger switching may be uncovered. Such information could be useful for the design of experiments in vivo. As a perspective, such models may help optimize specific exercise regimens and, given sufficient data, optimize them individually for each athlete.

However, there is not enough data to build a complete model. There are too few articles on the activity of kinases and phosphatases during exercise with sufficient time resolution. There is too little data on the substrate specificity of kinases and protein phosphatases. The complete isoform-specific interactome remains to be studied.

## Figures and Tables

**Figure 1 ijms-24-11223-f001:**
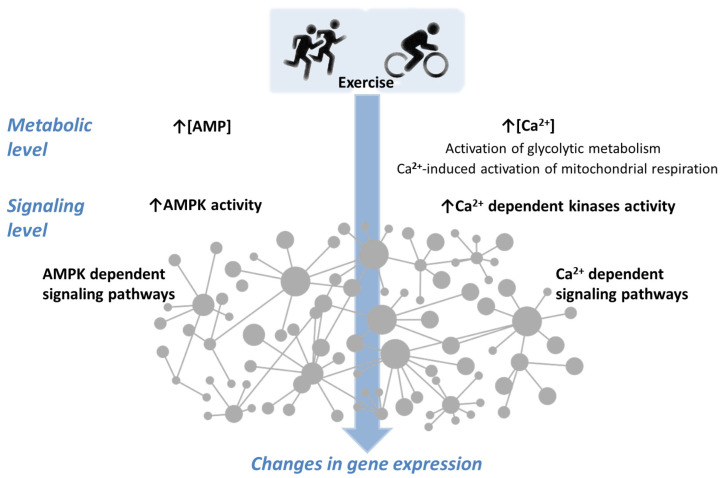
A simplified scheme of Ca^2+^- and AMPK-dependent signaling. Exercise causes numerous changes at the metabolic level. An increase in [AMP] and [Ca^2+^] concentrations, indicated by an arrow (↑), activates coupled signaling pathways that form a complex network with numerous feedback loops and protein–protein interactions. This leads to changes in gene expression.

**Figure 2 ijms-24-11223-f002:**
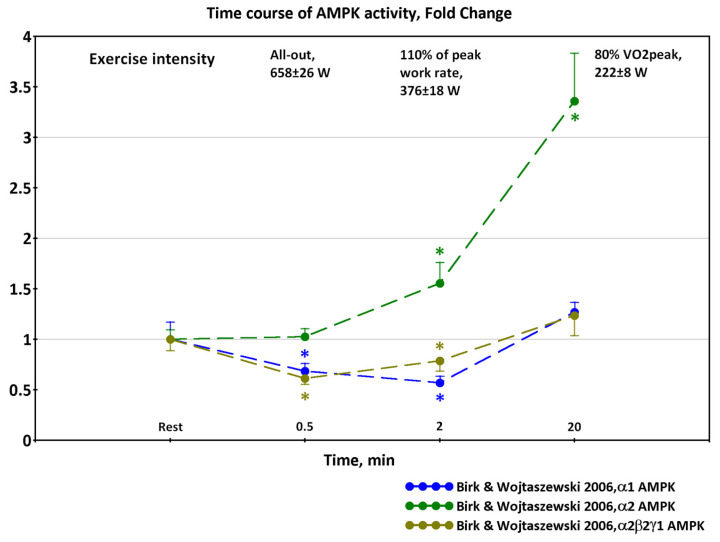
AMPK subunit-associated activity in response to exercise: trends based on data from [16]. A decrease in α1 AMPK activity after high-intensity sprinting exercise. Significant: (*).

**Figure 3 ijms-24-11223-f003:**
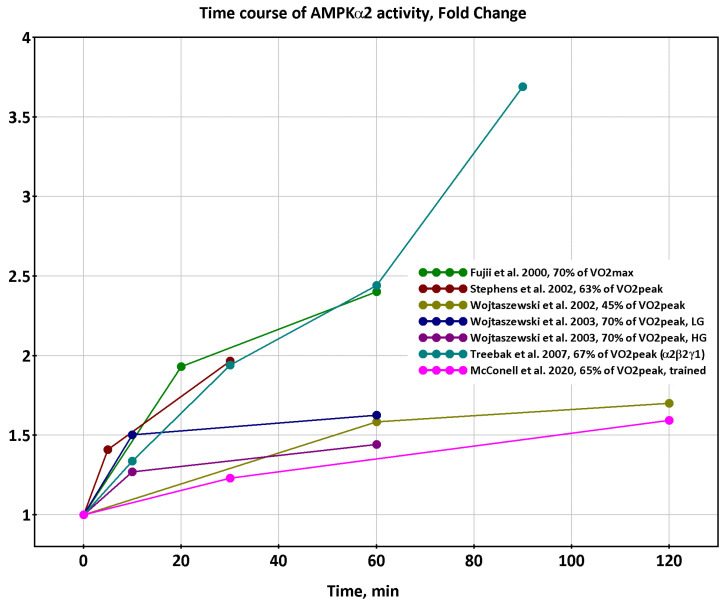
The progressive increase of AMPK activity (trends). Based on data from [12,15,20,23,25,26]. Axes X and Y indicate the time of the corresponding exercise and the fold increase of the AMPK activity compared to the pre-exercise state, respectively.

**Figure 4 ijms-24-11223-f004:**
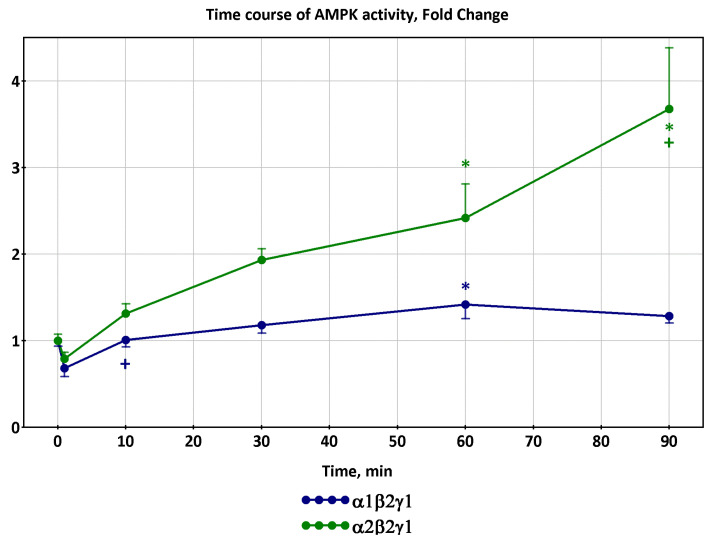
A decrease in the activity of various isoforms of AMPK at exercise onset (cycling exercise for 90 min at 67% VO_2peak_), trends based on data from [12]. Significant: (*), different from the preceding time point: (+).

**Figure 5 ijms-24-11223-f005:**
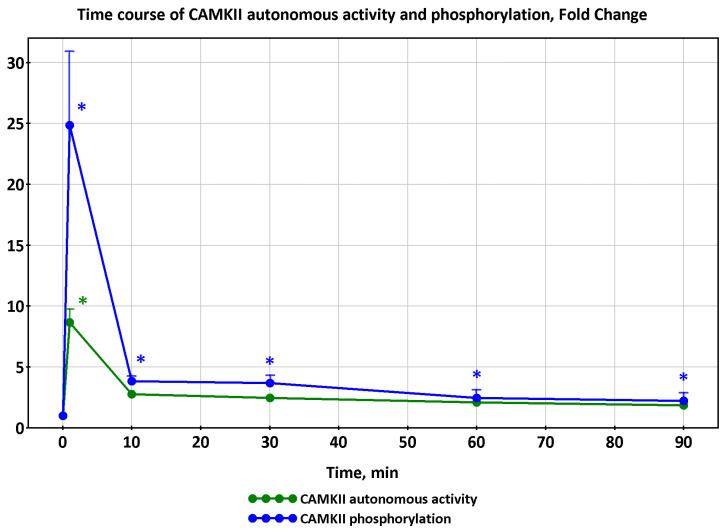
Effect of exercise duration on skeletal muscle CaMKII autonomous activity and phosphorylation, based on data from [27]. Significant: (*).

**Figure 6 ijms-24-11223-f006:**
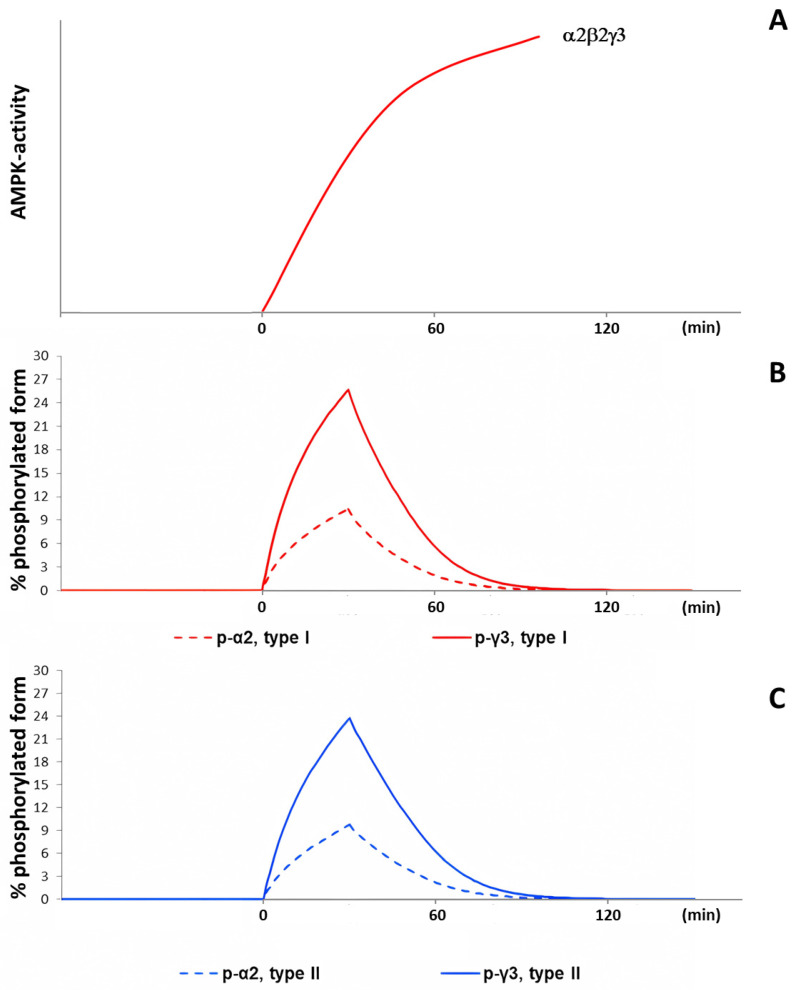
The nonlinear increase of AMPK activity during moderate-intensity exercise. (**A**) The increase of α2 AMPK activity, based on data from [24]. (**B**,**C**) Simulation results for the percentage of all α2 phosphorylated proteins (dashed) and of the phosphorylated γ3 heterotrimers (solid) in type I fibers and type II fibers, respectively, as an indirect measure of AMPK activity.

**Figure 7 ijms-24-11223-f007:**
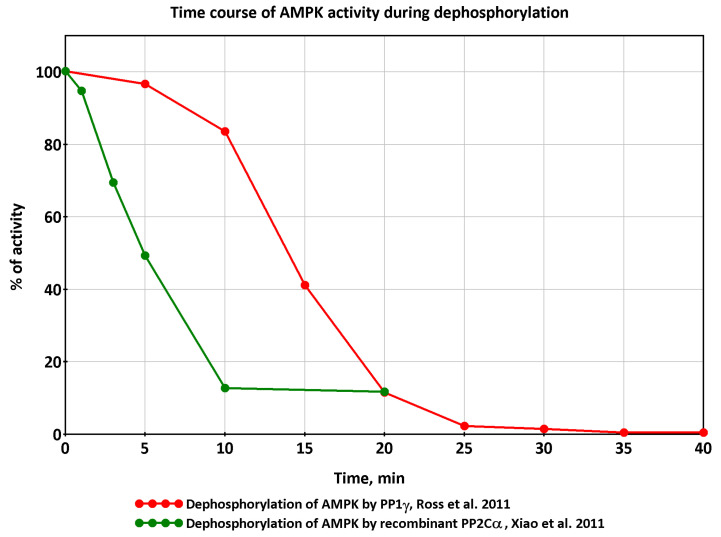
The activity of AMPK (α2β2γ1) that was phosphorylated by CaMKKβ (also known as CaMKK2) with subsequent incubation with recombinant protein phosphatase 1γ (PP1γ) or recombinant protein phosphatase 2Cα (PP2Cα), based on data from [38,39].

**Figure 8 ijms-24-11223-f008:**
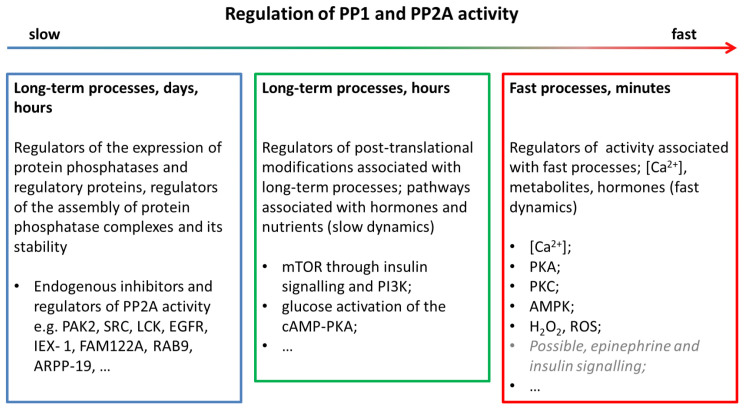
An approximate distribution of protein phosphatase’s regulatory factors in three time domains. For distribution, the dynamics of the response after stimulation, shown in the figures in the original papers, were used. To model transient processes during exercise, factors from the “fast” group were mainly considered as candidates.

**Figure 9 ijms-24-11223-f009:**
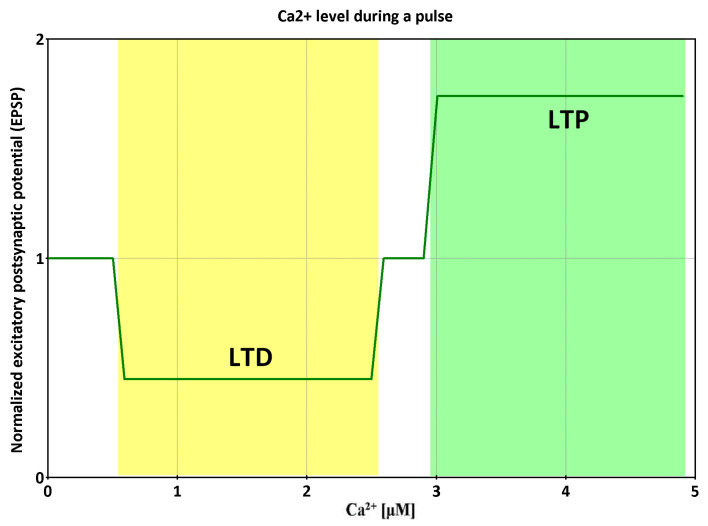
Coupled kinase and phosphatase switches can produce the tristability required for long-term potentiation (LTP) and long-term depression (LTD) of synapses, based on data from [45]. Dependence of the sign of synaptic modification on Ca^2+^ level during stimulation.

**Figure 10 ijms-24-11223-f010:**
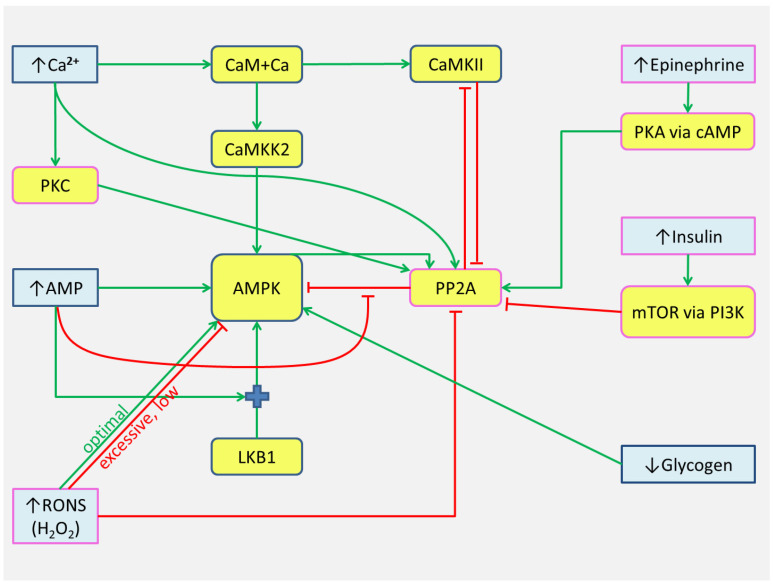
Proposed trigger interactions of kinases/protein phosphatases in the human skeletal muscle. Green lines indicate activation processes, while red ones designate inhibition. Objects with a blue border already exist in a recently developed model [3], while objects with a pink border are still not presented in the modular model. Objects with yellow shading designate kinases and phosphatases, while blue ones mean metabolites and hormones. The figure shows the main factors that change during exercise and rest periods between exercise bouts that can affect thresholds for switching potential triggers. Increase of concentrations is indicated by an arrow (↑), and decrease is indicated by an arrow (↓). Abbreviations: PKA, protein kinase A; mTOR, mammalian target of rapamycin; PI3K, phosphoinositide 3-kinases.

**Figure 11 ijms-24-11223-f011:**
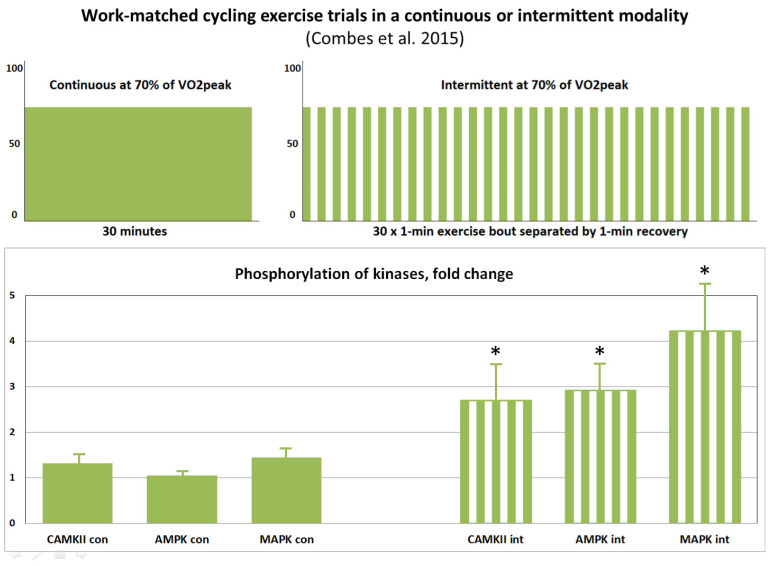
The effect of exercise modality on phosphorylation of CaMKII, AMPK, and p38 mitogen-activated protein kinase (p38-MAPK) proteins immediately after (+0 h) isocaloric exercises. Based on data from [92]. The phosphorylation of CaMKII, AMPK, and p38-MAPK markedly increases after intermittent exercise only. Significant: (*).

**Figure 12 ijms-24-11223-f012:**
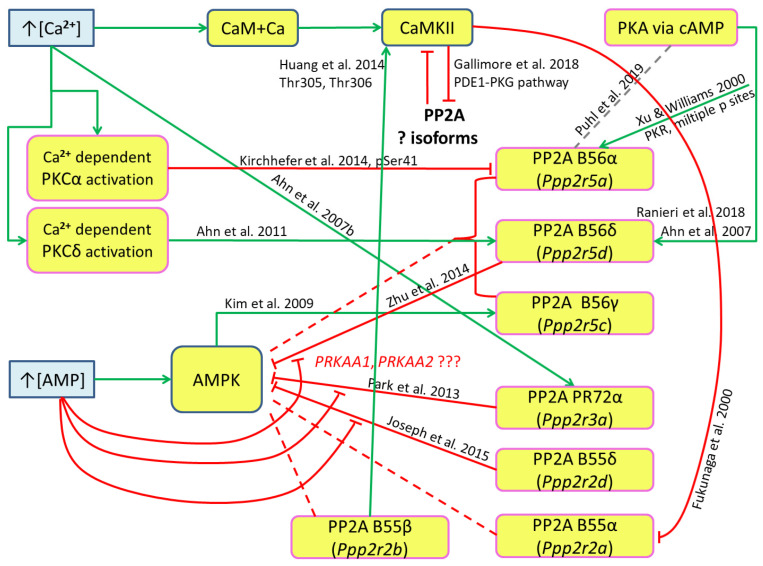
Proposed trigger interactions of kinases/protein phosphatases in human skeletal muscle in the context of the substrate specificity of protein kinases and protein phosphatases. Isoforms of protein phosphatase 2A are represented by the type of its regulatory subunit. Green lines indicate activation processes, while red ones designate inhibition. Other designations are the same as in Figure 10. Solid lines represent confirmed interactions using published data [48,49,51,62,89,90,118,119,120,121,122,123,124,125]. Dashed lines represent possible interactions. Increase of concentrations is indicated by an arrow (↑). Long green arrows indicate stimulation of activity. Question marks (?) indicate missing required data.

## Data Availability

Not applicable.

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
