# Peer review of "Numerous Trigger-like Interactions of Kinases/Protein Phosphatases in Human Skeletal Muscles Can Underlie Transient Processes in Activation of Signaling Pathways during Exercise"

_ijms, 2023, doi:10.3390/ijms241311223_

Round 1
Reviewer 1 Report
Numerous trigger-like interactions of kinases/protein phosphatases in human skeletal muscles can underlie transient processes in activation of signaling pathways during exercise
by Alexander Yu. Vertyshev , Ilya R. Akberdin 2,3,4*, Fedor A. Kolpakov
This review analyzes the internal processes occurring during exercise that initiate subsequent adaptation. During exercise in fact, muscle cells undergo a series of metabolic events that trigger downstream signaling pathways and induce the expression of many genes in working muscle fibers. Several studies analysed the dependence of changes in the activity of AMPK, one of the mediators of cellular signaling pathways, on the duration and intensity of single exercises. In particular, this review intends to summarize the findings on changes in the activity of AMPK, CamKII, and other components of the signaling pathway in skeletal muscles during exercise. The Authors hypothesized that the observed changes in AMPK activity may be largely related to metabolic and signaling transients rather than exercise intensity per se. The proposed hypothesis aims to reinterpret the data obtained in many studies and make assumptions about ways to optimize training regimens in the future.
I found this review difficult to read, and scarcely focused. I often get lost in the text, that sounds to me not logically, exhaustively and well explained. Maybe it is my fault, but I have not comprehended the aim of this paper. I include some criticisms below but I cannot endorse the publication of this article.
Major criticisms
-
page 2: “The data below suggest the existence of a number of specific transients during the activation of Ca 2+ - and AMP dependent signaling pathways in skeletal muscle that may lead to unexpected responses, such as a decrease in the activity of certain AMPK isoforms in response to short-term exercise”. It is not clear to me what the “specific transients” are. It should be clearly specified since it is a main topic of this paper.
-
There is no data on what transients exist during the variable intensity training and competitions. For example, during various interval training exercises or when moving along a course with uphills, downhills and flat terrains.
-
page 5 of 25: Authors claim: “If the biopsy was taken more often, then signs of transient processes were observed. At the beginning of the exercise, a downward trend in the activity of α1β2γ1 and α2β2γ1 isoforms of AMPK was observed Treebak et al. 2007 [17] as well as slow rise of α2β2γ3 activity. Then the AMPK activity was increased during the progression of exercise (Figure 3)”. In my opinion this paragraph is not sufficiently clear. The choice of the biopsy here sounds pretty new, and not introduced before. I think that authors should better explain the concepts without the necessity for the reader to go back to the appropriate reference.
-
page 5 : Authors claim: “There is no data on what transients exist during the variable intensity training and competitions. For example, during various interval training exercises or when moving along a course with uphills, downhills and flat terrains”. It is not clear to me if all the reported data and shown in the Figures, come from biopsies. In that case I think it is quite hard to get voluntary biopsies from subjects executing that kind of training. Please discuss it.
-
page 11. Para 4.4: “An exercise induces the increase of adrenaline: WHERE ? please define
and noradrenaline concentrations and this increase depends on the duration, intensity, and type of exercise. β-adrenergic receptor stimulation activates cAMP signaling cascade and may indirectly promote AMPK activity Aslam & Ladilov 2022 [83]. Increasing the concentration of cAMP HOW ? may promote AMPK activity by enhancement of the AMP: ATP ratio
-
Figures are often hard to read both for the poor resolution and lack of explanation in the legend
Minor criticisms
Abstract:
-
last line : makes instead of make
Introduction:
-
Carefully read the Manuscript preparation rules about citations. Use [numbers ] to indicate the reference but eliminate Authors name and year in the text
-
VO2max: please, use the correct subscript
-
Figure 1: exercise intensity (not intencity)
-
Figure 5 is of poor quality (numbers on X axes are hardly readable . The same for Y axes.) Title lacking in Y axes of panel A
-
carefully check PP2A and PPA2 throughout the text
-
Figure 7: Authors should explain in figure legend all the terms present in the graph
Author Response
We would like to thank the reviewers for their critical comments and helpful suggestions. Based on these comments and suggestions, we have made careful modifications to the original manuscript. The reviewer’s comments are shown in blue, followed by our responses in black in the attached file. The modifications made to the manuscript following the comments are marked in the revised manuscript.

Reviewer 2 Report
Vertyshev, Akberdin and Kolpakov review the extant data on the phosphorylation status/activity of AMPK and CaMKII in muscle tissue in response to various exercise regimes. The authors use the discrepancies between experimental data and their current best mathematical model as a springboard to explore some of the potential factors that may be missing from their model. By analogy with comparatively well-studied transient signalling processes in the central nervous system, they propose the existence of a ‘trigger-like’ network of feedback loops connecting AMPK and CamKII with their respective regulatory phosphatases and suggest this may be responsible for the generation of signalling transients during exercise onset. They summarise the role of additional factors, including hormones, glycogen stores and reactive oxygen and nitrogen species, and build these into their network model. Finally, they summarise remaining gaps, in particular the role of isoform specificity amongst the kinases and phosphatases in their network. It will be interesting to see these concepts developed into a working model and tested against experimental data.
I would recommend this review for publication once the authors have addressed the following points:
Major issues:
· Many of the sections are rather lengthy and would be easier to read if split into sub-sections as is the case in section 4.
· Please could the authors confirm that they have permission to re-use the data for their figures from the publications that the cite?
· The citation formatting is confusing – please use either [numbers] or (author + date), not both!
· The authors provide limited discussion on the likely error bars or statistical significance in the experimental data that they reproduce in their figures, nor do they mention the number of experimental subjects. This reader would appreciate at least a brief mention of these potential limitations either in the main text or the figure legends or both.
· In several places through the manuscript the authors use the phrase ‘enzymes, kinases and phosphatases’; kinases and phosphatases are two subsets of the larger set of enzymes. Please be more specific about the other types of enzymes you mean (perhaps metabolic enzymes?) or remove the word enzymes altogether.
· When the authors refer to the activity of the two kinases (AMPK and CaMKII) upon which they focus their attention, it would be helpful to clarify how this is measured in the different studies to which they refer. For example, is activity measured by looking at the phosphorylation status of the kinases’s substrates? Or is it rather measured by using the phosphorylation status of the kinases themselves as a proxy? If the measure of ‘activity’ is different amongst the different studies, might this explain some of the differences in the results obtained?
· Several statements in the manuscript appear to lack supporting references; these are (1) section 2, sentence 2, subsentence 1 (about changes in AMPK alpha2); (2) section 2, paragraph 7, sentence 2; (3) section 2, penultimate paragraph, no reference provided for ‘the current set of model equations’ – is this Akberdin, 2021 or Jensen 2009 or both, or some others?; (4) section 3, paragraph 7, sentence 3 (CamKII:PP2A ratios in muscle)
· The paragraph which introduced Figure 5 leaves this reader in some confusion. I think perhaps the authors mean: “Our current model (Akberdin et al, 2021) shows a nonlinear increase of AMPK activity (ON-kinetics) during moderate intensity exercise (Figure 5B,C), which is similar to that proposed by Jensen et al (Jensen et al, 2009) (Figure 5A).” However, it remains unclear to me whether Jensen et al base their graph on experimental data or on a mathematical model?
· Section 3 – the authors provide only a partial list of the published data sources that they use in their analysis stating ‘and many other articles’ – please either provide relevant reviews that summarise all these articles or find a way to list them completely, for example in an appendix.
· Section 3, paragraph 5 is rather complex, and would be easier to follow if illustrated with a suitable figure, either within this text or be referring out to a suitable review. I don’t think Figures 8 or 10 encompass all the details in this paragraph.
· Section 4 – it would be helpful to the reader if the authors were to refer to Figure 8 at relevant points within this section as this helps one to follow the complexities detailed in the subsections.
· Section 6, paragraph 4 – what do the authors mean by “The scheme of isoform-specific interactions (Figure 10) shows the absence of all necessary data”? To this reader Figure 10 shows that some of the data (dotted lines) are missing, but not all. This sentence is misleading – please reword to better capture the situation presented in Figure 10.
· Section 7 – I am confused by the sentence ‘There are not necessarily real triggers with multiple stable states, but even significant shifts in the balance of kinase/phosphatase activities that can lead to the observed dynamics of indicators.’ Should the word ‘even’ be perhaps ‘rather’ or instead’? or do I misunderstand your hypothesis completely?
Minor points (grammar, typographical errors, etc):
· There are some typographical and grammatical errors that merit attention – I have highlighted some of these below, however, a careful additional English edit would certainly be beneficial.
· The abstract is written in mixed tenses – please ensure a uniform use of either past or present tense throughout.
· I am not sure the wording of the final sentence of the abstract conveys the meaning the authors intended. A clearer wording might be similar to this “The proposed hypothesis allows for a reinterpretation of the experimental data available in the literature as well as for the generation of ideas to optimise future training regimens.”
· Throughout, please replace “at the opposite side” with ‘on the other hand’.
· Introduction, paragraph 1, final sentence – the word ‘complications’ is not necessary here.
· Section 2, paragraph 1; AMPK during exercise does not change (rather than ‘does not changes’)
· Figure 1 – misspelling of intensity.
· Please ensure that the characters for alpha, beta, etc are written in the correct font throughout the figures, I noticed issues in Figures 1, 2 and 3.
· In the second paragraph following Figure 3, I think the word ‘trend’ may be clearer than ‘phenomenon’ here, and, as both phenomenon and trend are singular, this should read ‘similar trend was observed’.
· Figure 5 – missing y-axis label for panel A; also it would be helpful for the reader to compare the figures if the x-axis were on the same scale for all three panels.
· In the paragraph immediately following Figure 5, the word ‘indistinguishable’ is incorrectly used – this should rather be ‘indiscernible’.
· Figure 6 – is the % necessary in the figure title? It doesn’t make sense to this reader.
· In the paragraph immediately following Figure 6, please replace the phrase “shown above” with the exact figure references. (is this Figure 5 A-C?)
· In the final paragraph of section 2, what do the authors mean by ‘AMPK activity (and others)’? – please expand on the ‘others’ or remove this phrase. Equally, in several additional places in the manuscript you use ‘others’ or ‘etc.’ – please be more specific rather than leaving the reader to speculate.
· I think the word interaction in the title to section 3, might be better as ‘interaction networks’?
· Figure 7 – please define the abbreviations in the figure legend (LTD, LTP) as well as the main text.
· Please define the abbreviation SOD in section 5.
Author Response

(The authors gave the same response as above.)

Reviewer 3 Report
Changes in AMPK activity may be due to metabolic and signaling transients rather than exercise intensity alone, with these transients occurring mainly in the first few minutes of exercise. These transients may be regulated by trigger-like kinase/protein phosphatase interactions, which are affected by factors such as Ca2+, metabolite concentration, and RONS. From a reviewer's point of view, the review explores these putative molecular mechanisms and suggests ways to optimize training regimens based on this understanding. I don't see any major issues with not accepting this manuscript in its current form. Only two minor issues to be considered:
1. it needs to be revised via a professional English editing service.
2. A figure in the introduction should be added describing AMPK and Ca2+ - dependent signaling
Author Response

(The authors gave the same response as above.)

Round 2
Reviewer 1 Report
Dear Authors
I must admit that the extensive revision of the MS has substantially improved the paper. However, I still have some points to raise:
Major
1) page 3/35. Authors claim that "the observed changes in AMPK activity may be largely related to metabolic and signalling transients rather than exercise intensity per se ". Could you please explain how exercise per se could change AMPK activity?
Minor:
1) is it requested by the Journal style to express references with names and numbers through the main text?
2) page 3/35. "Since there are many reviews Egan & Zierath 2013 [4], Gehlert et al. 2015 [5], Camera et al. 2016 [6], Kjøbsted et al. 2018 [7], McGee & Hargreaves 2020 [8] wrote a..." wrote does not sound correct.
3)page 3/35 spell check for printing and grammar errors (es can reflects, etc)
4) in the downloaded version of the MS I see three figures 10 and three figs 12 that look identical. Please check.
Author Response
We thank the reviewer again for his (her) comments and suggestions. We also provide our replies on the major and minor critics by points highlighting the reviewer’s comments by blue.
